# Overprecision increases subsequent surprise

**Don A. Moore** [1]*, **Derek Schatz**[2]

**1** University of California at Berkeley, Berkeley, CA, United States of America, **2** Accenture, PLC, San Francisco, CA, United States of America

* dm@berkeley.edu

## Abstract

Overconfident people should be surprised that they are so often wrong. Are they? Three studies examined the relationship between confidence and surprise in order to shed light on the psychology of overprecision in judgment. Participants reported ex-ante confidence in their beliefs, and after receiving accuracy feedback, they then reported ex-post surprise. Results show that more ex-ante confidence produces less ex-post surprise for correct answers; this relationship reverses for incorrect answers. However, this sensible pattern only holds for some measures of confidence; it fails for confidence-interval measures. The results can help explain the robust durability of overprecision in judgment.

**Data Availability Statement:** Pre-registrations (for Studies 2 and 3), materials, and data are available at http://osf.io/j5vpe/.

**Funding:** The author(s) received no specific funding for this work.

## Introduction

Overprecision is overconfidence in the accuracy of one's beliefs [1]. This excessive certainty is on display when we are too sure we are right [2], when we believe we can forecast others' behavior [3], when doctors are too certain of a favored diagnosis [4], or when managers issue excessively precise and inaccurate earnings forecasts [5]. Frequent feedback should help people calibrate their confidence; being routinely wrong should reduce confidence in the next forecast. However, the robustness and durability of overprecision suggests this corrective may be incomplete [6, 1]. When their expectations prove wrong, people ought to be surprised. Are they? In this paper, we ask how overconfidence contributes to subsequent surprise.

## Overprecision

There are innumerable ways in which overly certain beliefs can impair decisions. Overprecision contributes, for instance, to managers issuing too much debt when they underestimate the volatility of their firm's future [7], foregoing accounting corrections to their own forecasts of firm returns [8], and issuing excessively precise and inaccurate earnings forecasts [9, 5]. Those too sure of their beliefs will be more vulnerable to other biases, such as naïve realism [10] or the "false consensus" effect [11]. Excessive faith in their beliefs can also lead people to discount others' views [12], or even disparage others as biased [13]. Overprecision leads people to do too little to protect themselves against risks [14]. And overprecision blinds people to the need to consider other perspectives [15, 16]. These mistakes can have painful consequences for both individuals and organizations [17]. Given the costly consequences of overprecision, understanding its persistence is important.

**Competing interests:** The authors have declared
that no competing interests exist.

We compare different measures of overprecision. Researchers have most often used the
confidence interval paradigm employed by Alpert and Raiffa [18]. This method consistently
finds overprecision [19], though it is controversial. The two most common critiques consider
confidence intervals to be too difficult for participants to understand [20], and that people do
not naturally think about confidence in terms of confidence intervals [14]. We directly com-
pare the confidence interval method with more naturalistic measures of confidence in Study 2.
In addition to confidence intervals, participants also reported how confident they felt on a
7-point scale (from "*Not at all confident*" to "*Extremely confident*"). This scale lacks an objec-
tive benchmark, but might nevertheless do a better job capturing subjective feelings of confi-
dence. Study 3 invites participants to report how confident they are that the truth will be close
their best guess. We intentionally vary our measures of confidence across studies in order to
test the robustness of our results.

Study 1 asks participants how confident they are that the right answer is close to (within ten
pounds of) their guess. Likewise, Study 3 asks participants to estimate the probability that their
answer is correct. For these measures, overprecision would lead people to overestimate the
chance that their answer is correct. These methods are simpler to explain and has the addi-
tional benefit that it elicits probabilities, which seem to be easier for people to understand and
report than percentiles or confidence intervals [21]. We find that our key result is robust across
the different elicitations used in the three studies.

We should also note that our approach requires us to designate some answers as correct,
others as incorrect. Continuous response scales require setting the limbo bar at some particular
height, above which responses are too far from the truth, and count as incorrect. Choosing
where to set this bar will, to some extent, be arbitrary. While we are consistent within studies,
between studies we employ different scales and different criteria. The consistency of the effects
we document across these different accuracy criteria ought to contribute further to confidence
in the reliability of the effects we document.

## Surprise

Surprise is one of the basic emotions [22]. Its functional role is to highlight erroneous predic-
tions and to direct attention at the surprising stimulus [23, 24]. This function is sufficiently
universal that surprise has proven useful in studying beliefs and expectations among monkeys
and human infants [25, 26]. When something unexpected happens, it receives more attention
and longer gaze. The level of surprise one experiences and the duration of subsequent gaze is
positively correlated with the degree of difficulty making sense of an event [27]. Seeing some-
one levitate is more surprising than seeing them jump. The more unexpected an event, the
more intense the emotional reaction to it [28].

The more confident one is of one's beliefs, the more surprising it should be when those
beliefs turn out to be wrong. Given the ubiquity of overprecision in judgment, people should
be surprised regularly. Here, we seek to connect the literature on surprise with the literature
on overprecision in judgment. These literatures directly imply the hypothesis that predictions
made with greater confidence will produce greater surprise when they turn out to be wrong.
We test this prediction, and examine its consequences for the correction of subsequent confi-
dence in judgment.

The literature on hindsight bias provides reason to think that surprise might be insuffi-
ciently responsive to outcomes that ought to be surprising [29]. The hindsight bias documents
the tendency to incorrectly recall antecedent events so as to make outcomes as more predict-
able than they were. By selectively recalling outcome-consistent facts, people render those out-
comes less surprising than they would otherwise be [30]. The hindsight bias might therefore

inflate confidence ex-post and impede the potential for surprise to play a corrective role reducing confidence in subsequent predictions [31].

## Present research

We asked participants to report ex-ante confidence for a variety of judgments, and after receiving performance feedback they then reported ex-post surprise at the result. This research hones in on a key interaction, wherein the relationship between ex-ante confidence and ex-post surprise is moderated by whether one's answer is correct. We present three studies examining this relationship. Study 1 examines the relationships between confidence, correctness, and surprise for both self and others. Study 2 exogenously manipulates confidence and replicates the key finding from the first study. Study 2 uses a different manipulation of confidence and employs a repeated-measures design to explore the temporal relationship between confidence and surprise. This longitudinal design allows us to examine the effect of surprise on subsequent confidence. That is, Study 2 asks whether surprise induced by being confident and wrong has the appropriate corrective effect of reducing subsequent confidence. Finally, Study 3 considers lay predictions regarding how surprised people believe they or others should be. Across the three studies, we employ a variety of different measures of belief precision and subsequent surprise, thereby testing their relationships with each other.

We report how we determined our sample size, all data exclusions, all conditions, and all measures. Pre-registrations (for Studies 2 and 3), materials, and data are available: http://osf.io/j5vpe/. The studies we report were reviewed and approved by the Committee for the Protection of Human Subjects at the University of California at Berkeley. All participants provided informed consent.

## Study 1: Accuracy and surprise

Study 1 examined the basic relationship between surprise and overconfidence. We hypothesized that both confidence and correctness would positively predict surprise. We also computed a measure of absolute distance from the true answer as an additional predictor on surprise. We expected higher ex-ante confidence would produce lower ex-post surprise for correct judgments, and for incorrect judgments higher ex-ante confidence would produce greater ex-post surprise. Study 1 seeks the antecedents of surprise, by examining the effect of confidence, correctness, and distance from the truth on subsequent surprise.

### Method

We obtained complete responses from 446 Amazon Mechanical Turk workers [32] who were located in the United States and with at least 95% approval rates. (An additional 141 failed to complete the survey, most because they failed an attention check.) We selected our sample size ex-ante, guided by the results of several pilot studies, and did not analyze the data until collection was complete.

After obtaining consent, the survey presented instructions followed by an attention check. The survey ejected those who failed the attention check. Remaining participants then saw five photographs of strangers. These photographs, showing the target individuals wearing exactly the clothing they were wearing when they were weighed, showed their full bodies. Participants had to guess the weight and respond to the question, "How likely is it that you have estimated the weight correctly within 10 pounds? Move the slider to indicate your level of confidence (0 means no chance, 100 means absolutely certain)." After each of the five images, participants saw the true weight of each target in turn, followed by truthful feedback on whether or not

their estimate fell within ten pounds of the actual weight. Following each round of feedback, participants reported their surprise on a scale of 0–100.

We did not collect any demographic information on participants in Study 1.

A survey programming error led to 29 rounds (1.3% of the 2230 total) for 24 participants in which the survey informed them that their answers were correct when they were not. The results below exclude these problematic rounds, but results are not meaningfully different when they are included.

## Results and discussion

Results reveal participants to be overprecise: The average hit rate for estimates within ten pounds of the true weight was 36%, yet average confidence was 66% ($SD$ = 20.8%). Given the repeated measures design, we employed a multilevel regression model with random slopes. We measured distance as the absolute value of the difference between the participants' estimates and the true weight of the person in the photograph.

We conducted a multilevel regression predicting surprise from absolute distance, correctness, confidence, and the interaction between correctness and confidence, all nested at the individual level. The results reveal that absolute distance did indeed positively predict participants' reported surprise, $B$ = .25, $t$ = 6.23, $p <$ .001. There were also significant main effects of confidence and correctness on surprise. Greater ex-ante confidence predicted greater ex-post surprise, $B$ = .61, $t$ = 17.24, $p <$ .001. Being correct was also associated with greater surprise, $B$ = 57.02, $t$ = 15.26, $p <$ .001. These main effects are qualified by a significant confidence-correctness interaction wherein greater ex-ante confidence was associated with lower ex-post surprise when participants learned their guesses were correct, $B$ = -1.27, $t$ = -24.14, $p <$ .001.

We next compared correct and incorrect answers by analyzing them with separate multilevel regressions. When participants were incorrect, more ex-ante confidence increased ex-post surprise, $B$ = .61, $t$ = 16.70, $p <$ .001, and greater absolute distance from the truth also produced greater surprise, $B$ = .24, $t$ = 5.79, $p <$ .001. When participants were correct, confidence predicted less surprise, $B$ = -.63, $t$ = -14.85, $p <$ .001. However, distance from the truth was not significant, $B$ = .09, $t$ = .42, $p$ = .67.

These results show that being confident and wrong is associated with greater surprise. However, this study is correlational in nature, leaving open the possibility that some third variable leads to both confidence and surprise. The next study employs an alternative measure of confidence and provides an experimental test of the effects of confidence on surprise using a longitudinal design. As we shall see, Studies 2 and 3 replicate the key results of Study 1.

## Study 2: Surprise over time

Study 2 sought to replicate Study 1's results in a different domain while expanding our scope to two new facets of investigation. Specifically, we examined how aleatory or epistemic questions could affect the relationship between confidence and surprise. Because people are more confident when uncertainty is epistemic than when it is aleatory [33], this manipulation served as an experimental manipulation of confidence.

Ten questions were aleatory in nature, where the answer could not be known beforehand: What number ball would be drawn from the bingo cage? The other ten weight-guessing questions were epistemic in nature, with answers that participants could conceivably know, given sufficient skill at weight-guessing.

Study 2 assessed confidence by asking participants to report their confidence on a seven-point scale. In addition, participants reported a 50% confidence interval—the interquartile range, demarcated by the 25th and 75th percentiles of their subjective probability distribution.

Conceptually, these different measures all assess subjective certainty—the concentration of a subjective probability distribution. Greater certainty is reflected in an SPD more tightly concentrated around the best guess. They are useful tools for testing the degree to which the effect we identify depends on the specifics of the way confidence is elicited and generalizing our results.

In addition, Study 2 sought to examine the temporal dynamics of confidence and surprise. Does experience help people adjust their expectations and better calibrate their confidence? Does it reduce surprise? We provided our participants with immediate feedback and measured the consequences of that feedback (reported surprise) on their subsequent reports of confidence.

## Method

Our pre-registered research plan called for 115 participants recruited via Amazon Mechanical Turk, restricted to workers in the United States with at least a 90% approval rate. We based this number on the effect size from a previous study (available in our supplemental online materials), $f^2$ = .036, and power of 90%. Participants averaged 35 years old (SD = 11.2 years). The majority of participants (74%) identified as white, 8% as Asian, 11% as African American, 6% as Hispanic, and 1% as Native American. Most (85%) had at least some college experience and 56% had completed at least a four-year college degree. The majority (63%) identified as female.

Study 2 asked each participant for 20 estimates; half aleatory and half epistemic. We randomized, between subjects, both the order of these two question blocks as well as the order of questions within each block. Our operationalization of aleatory questions came in the form of drawing bingo balls numbered from 1 to 100. Our epistemic questions asked participants to guess weights from photographs (as in Study 1). For each estimate, we asked participants to provide a best guess of the answer, as well as high and low bounds to establish a confidence interval: first, "a number so **low** that you believe there is only a 25% chance the true weight [or actual bingo ball] falls below it" and also "a number so **high** that you believe there is only a 25% chance the true weight [or actual bingo ball] falls above it." These interquartile ranges also defined which answers were considered 'correct' (i.e. whether the true answer was inside or outside of their self-created 'confidence' interval). Participants provided three responses per round: a best guess plus high and low bounds, at the 25th and 75th percentiles, to form a 50% confidence interval. We used a standardized measure of the size of participants' confidence intervals (their high estimate minus their low estimate divided by their 'best guess' estimate) as a confidence measure. In addition, we asked, "How confident are you that the actual bingo ball number about to be pulled will fall within this range?" on a scale from 1 (*Not at all confident*) to 7 (*Extremely confident*).

In addition, participants reported confidence on a 1 to 7 scale (from "*Not at all confident*" to "*Extremely confident*"). After learning the right answer, participants reported surprise on a 1–7 scale (from "*Not at all surprised*" to "*Extremely surprised*").

## Results and discussion

Hit rates between the 25th and 75th percentiles were similar for aleatory (65.83%) and epistemic questions (68.78%), $t(114)$ = -1.51, $p$ = .13. These hit rates suggest that participants were actually underprecise for both bingo ball questions and weight guessing questions, as both hit rates are greater than the 50% expected to hit in the interquartile range. However, the bingo rounds produced lower feelings of confidence ($M$ = 4.82, $SD$ = 1.85) than did the weight rounds ($M$ = 5.42, $SD$ = 1.38), $t(114)$ = -8.83, $p < .001$.

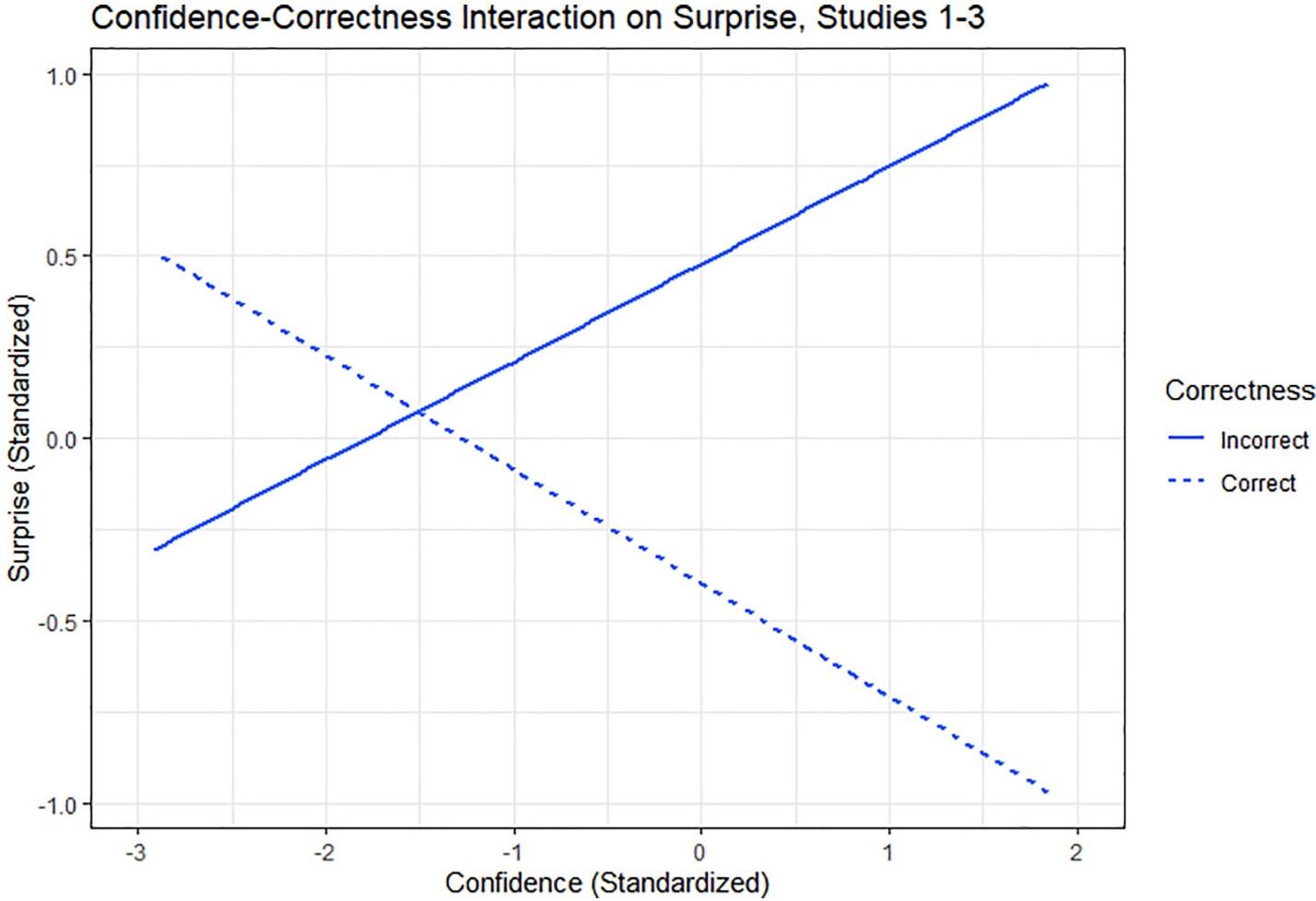

**Fig 1. The aggregated standardized interaction effect of confidence and correct answers on reported ex-post surprise, for Studies 1–3.** Measures are standardized by z-scoring using the grand mean within each study, then aggregating.

The effects of confidence and correctness on surprise replicate the results of Study 1: when incorrect, high confidence predicted *higher* ex-post surprise; though when correct, higher confidence predicted *lower* ex-post surprise. This result emerges from a multilevel regression predicting surprise with question type, correctness, distance, confidence, and a correctness-confidence interaction. There was a main effect of question type: the epistemic weight-guessing questions elicited greater surprise ($M = 3.40$, $SD = 1.91$) than the aleatory bingo ball questions ($M = 2.71$, $SD = 2.15$), $B = .63$, $t = 9.63$, $p < .001$. This analysis also replicated the confidence-correctness interaction from previous studies, $B = -.96$, $t = -23.52$, $p < .001$. This interaction arises because confidence increased surprise only following incorrect answers. Subsetting on correctness identifies the negative relationship between confidence and surprise when correct, $B = -.34$, $t = -12.89$, $p < .001$, and the positive relationship between confidence and surprise when incorrect, $B = .43$, $t = 9.37$, $p < .001$. See Fig 1.

However, our other measure of confidence (confidence interval size) produces different results. The same multilevel model predicting surprise with question type, correctness, distance, and confidence (from interval size), produces a correctness-confidence interaction, $B = -.96$, $t = 5.16$, $p < .001$. Again to seek clearer insights, we ran this new model on data subsetted by correctness. The negative relationship between confidence and surprise for correct answers

is weakened to nonsignificance, $B = -.02$ $t = -1.55$, $p = .12$. The positive relationship we found previously between confidence and surprise when incorrect is nearly nonexistent, $B = .04$, $t = 0.59$, $p = .56$.

The relationship between confidence interval size and self-reported confidence is not intuitive for participants. Confidence interval size was correlated with self-reported confidence at $r = .28$, $p < .001$. This suggests that as intervals grew (implying lower confidence), self-reported confidence also increased—a contradictory pattern. Puzzlingly, this correlation between CI size and scale confidence only exists with incorrect answers, $r = .30$, $p < .001$, the correlation is nonsignificant for correct answers, $r = .05$, n.s. This is intriguing as both measures of confidence are reported prior to participants learning the correct answer.

This study's repeated-measures design allowed us to test the effect of surprise on confidence over time. We employed a lagged regression to see how confidence is predicted by correctness at $t-1$ (i.e., how correctness in any given round predicts confidence in the round immediately following). Lagged correctness at $t-1$ significantly predicted confidence at time $t$, $B = .25$, $t = 4.12$, p $< .001$. In other words, participants expanded their confidence intervals after having been wrong. Did reported surprise impact subsequent confidence similar to how correctness affected subsequent confidence? Adding lagged surprise at $t-1$ into the same model showed no significant effect of lagged surprise on confidence, $B = .002$, $t = 0.13$, $p = .89$. This result suggests a profound failure of the functional role of surprise: it did not reduce subsequent confidence. The inclusion of lagged surprise also left the main effect of lagged correctness on surprise nearly unchanged, $B = .26$, $t = 3.78$, $p < .001$.

Results of Study 2 introduce important caveats to our key interaction between confidence and correctness on reported surprise. The results reinforce the importance of the method one uses to elicit confidence, as our key results largely disappear using the confidence-interval elicitation. This might be attributable to the difficulties people have setting confidence intervals. For instance, people set 50% confidence about as wide as they set 98% confidence intervals, despite the fact that 98% confidence intervals should be much wider [34]. These results add to our skepticism of confidence-interval measures and the degree to which they effectively capture subjective feelings of confidence. However, there is another possibility: Because two effects may have cancelled each other out: (1) more confident people were more surprised at being wrong but (2) people who gave wider intervals (i.e. those less confident) were more surprised that their larger intervals did *not* contain the right answer. This ambiguity interpreting confidence interval measures led us to abandon confidence interval elicitations in Study 3.

## Study 3: Predicting surprise

Study 3 asks whether people are as surprised as they *should* be. Lacking an normative benchmark for degree of surprise, we pre-registered a plan that asked a subset of participants how surprised they thought they ought to be. Anticipating that simply asking the question might influence subsequent reports, the study randomly assigned half the participants to a prediction condition which elicited predictions of surprise. The other half of participants, in the control condition, made no such predictions. Study 3 again employed both aleatory and epistemic judgments.

In choosing which tasks to use, we noted that Study 2 found surprise to be lower for bingo balls than weight-guessing. Since the distribution of bingo balls is uniform, all numbers are equally likely and there is little reason to be surprised by any particular outcome, potentially contributing to diminished "surprisingness." Therefore, Study 3 replaces bingo ball draws with a set of ten coin flips, which has a single-peaked distribution of outcomes and the potential for truly surprising outcomes (such as ten flips all coming up tails).

## Method

A power analysis of the prior studies, in which people reported ex ante surprise, providing average effect sizes of $f^2 = .036$ to $f^2 = .024$ and a recommended sample size of 105. Wary of losing power from subsetting the data in testing our hypotheses, we pre-registered a sample size of 150 and ended up with 151 participants (64% female). We restricted participation to Amazon Mechanical Turk workers in the United States with at least a 90% approval rate and 50 approved work assignments. Participants averaged 36.4 years old (SD = 11.4 years). The majority of participants (78%) identified as white, 9% as Asian, 5% as African American, 5% as Hispanic, 2% as Native American, and 1% as other. Most (84%) had at least some college experience and 50% had completed at least a four-year college degree.

Each survey informed participants that there would be two trial blocks: ten rounds of ten coin flips each and ten rounds of guessing individuals' weights from images. For each of the two blocks (which were presented in a random order) participants provided a best guess estimate and reported their confidence in that estimate (on a 1–100 scale with endpoints labeled "Not at all confident" and "Extremely confident"). In the coin-flip rounds, the question read "How confident are you that the actual number of heads for this next round will be within 1 of your of your guess?" In the weight-guessing rounds, the question read "How confident are you that the actual answer for this image will be within 10 pounds of your of your guess?" Following each round, we truthfully informed participants whether they had answered correctly. Then participants responded to the question, "How surprised are you at the actual number of heads for this round?" or "How surprised are you that the actual person's weight is [x] pounds?" (on a 1–7 scale with endpoints labeled "Not at all surprised" and "Extremely surprised"). Answers for coin flips counted as correct if they were within one head (out of ten) of the actual outcome. Weight guesses counted as correct if they were within ten pounds of the truth.

We assigned participants to one of two between-subjects conditions: a control condition and a prediction condition, where, in each round, in addition to the procedure described above, participants predicted how surprised they would be if their answer was right and if it was wrong:

"How surprised would you be if your answer for this next round was *correct*?" and

"How surprised would you be if your answer for this next round was *wrong*?" They responded on a 1 to 7 scale.

## Results and discussion

On average, participants were overconfident. They report being 55.5% confident on average, but they are only right 50.3% of the time, one-sample $t(2804) = 12.1$, $p < .001$. This confidence declines with experience. It starts at 60.2% in Round 1 and declines to 52.8% in Round 20. A linear regression predicting confidence with round number, coin/weight, and fixed effects for subject produces a strong effect of round, $B = -.59$, $t = -5.08$, $p < .001$. There is no significant difference in expressed confidence between weight and coin rounds, $B = -1.55$, $t = -1.16$, $p = .25$. This lack of a difference between weight and coin rounds is remarkable given that participants' guesses are correct 61% of the time for coin rounds but only 39.8% of the time for weight rounds, $t(3038) = 11.83$, $p < .001$.

Were participants as surprised as they predicted they should be? In order to test this, we employed two paired t-tests to account for the repeated measures design, comparing (1) the predicted surprise for a correct answer to a subsequent correct answer's surprise, and (2) the predicted surprise for an incorrect answer to a subsequent incorrect answer's surprise. We see that participants reported less surprise ($M = 3.30$, $SD = 1.91$) than they predicted they would

($M$ = 3.78, $SD$ = 1.82) when correct, $t(712)$ = 7.63, $p < .001$. Conversely, they were more surprised ($M$ = 4.77, $SD$ = 1.82) than predicted ($M$ = 3.61, $SD$ = 1.74) when incorrect, $t(761)$ = -15.56, $p < .001$. These patterns are similar for both coin and weight rounds.

We also explored whether the mere act of prediction had an impact on one's subsequent surprise. We employed a repeated measures one-way ANOVA predicting ex-post surprise by condition, controlling for individual-level error. This analysis shows that condition significantly predicts ex-post surprise, $F(1, 2888)$ = 19.41, $p < .001$. A follow up hierarchical linear regression controlling for individual-level error shows that being in the prediction condition increased ex-post reported surprise, $B$ = 0.65, $t$ = 4.27, $p < .001$. Those who predicted their surprise wound up reporting more surprise ($M$ = 4.04, $SD$ = 2.01) than those who did not ($M$ = 3.39, $SD$ = 2.04).

In testing a replication of the lagged analysis from Study 2 we ran a similar lagged analysis predicting confidence with correctness at $t-1$. The analysis shows that when it was the only predictor, lagged correctness positively predicted subsequent confidence, $B$ = .05, $t$ = 3.29, $p < .005$. However, including lagged surprise wipes out the relationship, leaving only lagged surprise significant, $B$ = -0.10, $t$ = -4.72, $p < .001$, where more surprise at $t-1$ led to less confidence at $t$. This contrasts with the result from in Study 2. Nevertheless, results from Study 3 do indeed document the corrective effect by which surprise reduces subsequent confidence.

## General discussion

Our results show that ex-ante confidence and ex-post surprise are inextricably linked. Our primary finding, as shown in Fig 1 and Table 1, is that when people are correct, greater ex-ante confidence produces less ex-post surprise, whereas when they are incorrect, greater ex-ante confidence produces more ex-post surprise. We examine the psychology underlying these relationships and identify moderators that can either suppress or enhance their strength. Studies 1 and 2 establish the link between confidence and surprise, highlighting that correctness is a powerful moderator of the relationship. Studies 2 and 3 employ exogenous manipulations of confidence; their results replicate the correlational results of Study 1. Study 2 finds more

**Table 1. Means for confidence and surprise, conditional on correctness, in the three studies.**

|  | Ex-ante confidence | Ex-post surprise | Confidence-surprise correlation (r) |
|---|---|---|---|
| Study 1 |  |  |  |
| Weights–right | 66.3% (20.7%) | 28.6 (28.1) | -.47*** |
| Weights–wrong | 65.7% (20.5%) | 60.6 (28.9) | .45*** |
| Study 2 |  |  |  |
| Bingo–right | 5.23 (1.82) | 2.40 (1.75) | -.43*** |
| Bingo–wrong | 4.03 (1.66) | 3.31 (2.06) | .45*** |
| Weights–right | 5.50 (1.38) | 2.42 (1.61) | -.38*** |
| Weights–wrong | 5.25 (1.37) | 5.57 (1.52) | .11* |
| Study 3 |  |  |  |
| Coins–right | 55.3% (25.2%) | 2.62 (1.73) | -.18*** |
| Coins–wrong | 54.9% (25.9%) | 4.20 (2.00) | .21*** |
| Weights–right | 49.7% (26.0%) | 3.17 (1.93) | -.37*** |
| Weights—wrong | 46.9% (27.1%) | 4.87 (1.71) | .09 |

Standard deviations in parentheses. The right-most column shows the correlation (r) between confidence and surprise.

*p < .05

***p < .001

powerful confidence-correctness interaction effects on surprise for epistemic questions than for aleatory, consistent with the notion that feeling personally accountable for knowing or not knowing the answer increases the intensity of emotional reactions to being right or wrong. Study 3 finds that people are more surprised about being wrong than they expect to be.

We must note the idiosyncratic nature of the laboratory contexts in which we conducted our experiments. They were designed more for experimental control and causal identification than for their similarity to any particular field context. It is possible that the higher stakes associated with some life events might facilitate learning. After eating a surprisingly spicy pepper, people's confidence taking the next pepper might be shaken. On the other hand, religious adherents' certainty does not always adjust downward when prophesied events fail to occur [35, 1]. As stakes increase, it is possible that ego-protection may impede the learning process. Testing the effect of different incentives is a potentially fruitful avenue for future research. Such research will have to grapple with the complex array of incentives, psychological, pecuniary, and interpersonal, attached to judgments, forecasts, social displays, and self-perception.

What of the utility of surprise? Surprise, as one of the basic emotions, serves a powerful and fundamental role stimulating curiosity and directing attention [22, 24]. If surprise reflects prediction error, individuals should seek to maximize accuracy and minimize surprise [36]. This implies that surprise should lead people to reduce their subsequent confidence. Our results suggest that surprise does not always play this functional role, or that it is difficult to measure consistently. Future research should examine the conditions under which surprise has a corrective effect on subsequent confidence. The results of Study 3 suggest that anticipating what would constitute a surprising outcome may help reduce the degree to which the hindsight bias allows people to persuade themselves that they knew it all along (see [37, 38]). How quickly does this effect decay and what possible moderators could increase the calibrating power and longevity of feedback on subsequent confidence? Could incorrect answers in epistemic domains more central to one's self-concept 'stick' for a longer period of time, forcing one's re-evaluation of their believed expertise? Or could the opposite be the case, where the incorrect answer is considered anomalous and the sense of expertise persists? These are questions for future research.

We aspired to measure the effects of overprecision on surprise. In recording participants' ex-ante confidence, their correctness, and their ex-post surprise, we document consistent evidence suggesting that people expect to be correct. If they go into a decision with confidence, they are more surprised to be incorrect, and less surprised when correct. We believe these results do more than underscore precision in judgment. Rather, this research approaches the topic with a new paradigm that serves to reveal another layer in the scientific understanding of the psychology of confidence and precision in judgment.

## Author Contributions

**Conceptualization:** Don A. Moore, Derek Schatz.

**Data curation:** Don A. Moore.

**Formal analysis:** Derek Schatz.

**Supervision:** Don A. Moore.

**Writing – original draft:** Don A. Moore, Derek Schatz.

**Writing – review & editing:** Don A. Moore, Derek Schatz.

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
