## [Decision Letter · Decision Letter 0]

26 Feb 2020

PONE-D-19-33924

Overprecision Increases Subsequent Surprise

PLOS ONE

Dear Dr. Moore,

Thank you for submitting your manuscript to PLOS ONE. After careful consideration, we feel that it has merit but does not fully meet PLOS ONE’s publication criteria as it currently stands. Therefore, we invite you to submit a revised version of the manuscript that addresses the points raised during the review process.

We would appreciate receiving your revised manuscript by Apr 11 2020 11:59PM. To enhance the reproducibility of your results, we recommend that if applicable you deposit your laboratory protocols in protocols.io, where a protocol can be assigned its own identifier (DOI) such that it can be cited independently in the future. For instructions see: http://journals.plos.org/plosone/s/submission-guidelines#loc-laboratory-protocols

We look forward to receiving your revised manuscript.

Kind regards,

Philipp D. Koellinger, Ph.D.

Academic Editor

PLOS ONE

Additional Editor Comments (if provided):

Both reviewers provided excellent, in-depth comments on your study. The internal validity of your results is obviously a "sine qua non", and many of the comments by R2 are essential in this regard. I also share R1's concerns about the external validity of your results, but this is a much more difficult goal to achieve. If additional data collection would be possible to address R1's concerns, that would be fantastic. At a minimum, please include a thorough discussion of the potential limits to the external validity of your results.

Journal Requirements:

Reviewers' comments:

Reviewer's Responses to Questions

**Comments to the Author**

1. Is the manuscript technically sound, and do the data support the conclusions?

Reviewer #1: Yes

Reviewer #2: Partly

2. Has the statistical analysis been performed appropriately and rigorously? 

Reviewer #1: Yes

Reviewer #2: I Don't Know

3. Have the authors made all data underlying the findings in their manuscript fully available?

Reviewer #1: Yes

Reviewer #2: Yes

4. Is the manuscript presented in an intelligible fashion and written in standard English?

Reviewer #1: Yes

Reviewer #2: Yes

5. Review Comments to the Author

Reviewer #1: Review for the paper entitled ‘Overprecision increases subsequent surprise’

Thank you for letting me review this exciting and relevant paper. I believe the authors have investigated a relevant topic in three rigorous studies. Their results show that overprecision increases surprise and reduces subsequent confidence. Although I think this research is important I have some concerns before the results can be published.

Major concerns:

1. My most important concern about the results that are reported in this paper is their external validity. All studies use abstract tasks to measure overprecision and I am not convinced that the results would hold for decisions that have personal consequences for the decision-maker. People in your studies had no incentive to guess the weight of people right. However, people who make overprecise predictions about their future income will experience personal consequences of this judgment. Similarly, managers who have to predict financial losses of their firm will probably experience personal consequences of a wrong judgment. Having said that, I also guess that people will put more thought into judgments that have personal consequences for them which will affect their subsequent surprise. Hence, before I can trust the presented results, I would like to see their external validity and see if this “sensible pattern” holds for (more) realistic settings.

2. You state that “When something unexpected happens, it receives more attention and longer gaze […].” As most people are uncertainty averse, they try to avoid or diminish uncertainty or the unexpected. Hence, I would argue that people would like to engage in deliberate consideration, especially for epistemic questions, to exactly avoid to be surprised for situations of personal relevance. In your weight guessing task, some people might not think systematically about cues that would give them the correct answer, but would just make guesses based on unspecific cues. Did you check participants’ time to give their answers or ask them about how they guessed the weight of the strangers?

3. Can you really compare the tasks of your studies? The probability distributions in the three tasks used is very different and I guess surprise is lower when you are correct in a coin flip compared to when you are correct in the bingo task or the weight guessing task. Similarly, you measure overprecision AND surprise differently. Why did you use 1-100 in Study 1 and 1 to 7 scales in Study 2 to measure suprise? Why did you not use the surprise subscale from PANAS-X? Related to that, your study design seems quite straightforward; is it possible that participants guessed the purpose of the study and give answers that conform with your hypotheses? Did you ask participants if they knew about the purpose of the study (and control for that)? Although I am no opponent of using MTurk for data collection, it would be interesting to describe your samples in a bit more detail (gender, age, ethnicity, profession etc.) and possibly also include some control variables in your analyses (at least in Study 1 because it is correlational).

4. I am missing some explanations why surprise does not reduce overprecision. Why should we care that overprecision leads to surprise if we don’t know which consequences surprise has for overprecision or any other (personally relevant) decision? It is possible that not surprise, but another emotion, for example, regret, could be a corrective to reduce overprecision. People may want to avoid feeling regret when finding out that their initial judgment was wrong and regret anticipation could reduce overprecision. This is just an hypothesis, but it would be interesting to test this (in a realistic setting).

Minor comments:

1. A more detailed overview of how overprecision is measured in extant research would be helpful in judging if you apply and compare the relevant measures for overprecision.

2. Did the photographs show the full body or only faces?

3. In Study 2, you used 10 questions per domain? Did you (quasi) randomize the questions or blocks of questions?

Reviewer #2: Overall

The idea that greater confidence will produce greater surprise (if incorrect) would seem to be a fairly intuitive thing. The manuscript in its current form doesn’t seem to build a very strong rationale for the research other than nobody else has done it before. Arguably, however, a number of studies in the hindsight bias literature have already demonstrated the relationship being tested in this paper (e.g., Ash, 2009; Choi & Nisbett, 2000; Pezzo, 2003). In these studies participants predict the likelihood of an outcome; those whose predictions were most “off” were found to be most surprised. Inherent in a likelihood judgment is a statement of confidence. If person A predicts a 90% chance that the home team will win, she is more confident than person B who predicts only a 60% chance. As such she will be more surprised if the team loses. The authors’ use of distance of participants’ estimate from the true value as a measure of confidence is consistent with this idea. (Note however, that hindsight bias per se can get convoluted and doesn’t always give the mirror image of surprise). In any case, the manuscript would benefit from an expanded introductory section on the extant literature on surprise, including some of the work by Rainer Reisenzein, mention the hindsight literature, and how this research moves beyond that, and should address theoretical differences between their different measures of confidence.

STUDY 1

Study 1 has some inconsistencies in its data reporting. The manuscript says N = 430, but t-tests give 451 degrees of freedom. The data set provided on the OSF website is a “tall” data set (typical of multilevel modeling) with what should be 5 rows per subject. However, the total number of rows (N = 2240) divided by 5 = 448. A quick check of unique IDs also yields 448. Also, 6 of the 2240 rows have missing data. Finally, the manuscript says that 37% of the guesses were correct (which is consistent with the OSF data set), but the supplemental material says 35%. These inconsistencies need to be cleared up.

A much bigger problem is that the column marked “correct” in the OSF data set is not consistent with the description in the method. The method says that a guess was marked correct if it was within 10lbs of the actual weight. Although the actual weights aren’t shown in the data set, we would expect the range of guesses that are correct (for any given photograph) to be 20lbs (i.e., +/- 10 lbs). Assuming that everyone saw the same photo in a given “round” (I confirmed this by examining the Qualtrics survey) here are the highest and lowest weight guesses for each of the five pictures. Remember, these are just from people marked correct in the OSF data set:

Round 1: Low guess = 150, High guess = 265

Round 2: Low guess = 120, High guess = 210

Round 3: Low guess = 95, High guess = 170

Round 4: Low guess = 120, High guess = 195

Round 5: Low guess = 105, High guess = 180

Clearly something is amiss here. No matter what the true weights are, these all have ranges much greater than 20lbs. As a double check, I inspected everyone, including those who were marked incorrect, and the ranges get even larger (as would be expected), and include some clear typos (e.g., weight estimates below 50lbs). I don’t see anything in the supplemental materials about cleaning any of this data.

I found the true weights in the Qualtrics survey and recalculated the correctness variable. Of the 2240 cases 35.9% were marked correct, but and most important, the original and the newly calculated “correct” columns are completely unrelated (kappa = .09).

The good news is that when you calculate the correlation between surprise and confidence (broken down by correctness, and using the proper “correct” variable), the simple bivariate correlations make a heck of a lot more sense:

r (incorrect) = .40 r (correct) = - 39. Exactly as you would expect.

STUDIES 2 and 3

The data sets for Studies 2 and 3 seem fine, near as I can tell (although I did not look closely). The method for all three studies should be fleshed out more (e.g., exact wording of the confidence interval question). I found myself often having to refer to the supplemental materials for basic information. The existing method section also has some unnecessary repetition that can be removed to make room for more of these details.

Overall, the presentation of the results is somewhat disorganized. A number of results are presented outside the context of the multilevel modeling and then again repeated within the modeling context (e.g., the main effect of question type). For example, at the top of p.10 a breakdown of the correct x confidence interaction is given before mentioning the interaction (and before mentioning MLM). Then the interaction is mentioned, and then essentially the same breakdown is given (with the exact same results and p-values repeated!). All three studies suffer from this problem. I think the manuscript would benefit from inclusion of a complete table with the results of all analyses in each study

Does distance from the true value (or confidence) treated as an individual difference variable (averaged across the 5 pictures for any given person) predict surprise at level 2? I think you can examine cross level interactions as well. Since you are using a random slopes approach (which I assume you tested for, and again should report) you might as well see what predicts the slope for any given person by examining level 2 predictors.

Study 2 uses a very different criterion for correctness (50% confidence interval) vs that of Studies 1 and 3 (“Is the given answer within +/- 10 lbs of the correct answer”), and the implications are never really discussed in the manuscript. In Study 3, the criterion for correct guessing of quarter flips seems a bit arbitrary. A “pilot” test could determine how many flips “off” most students would require before they considered their guess to be wrong. In general I wonder if the participants would agree with the definition of correctness and if there are individual differences. At least discuss this.

From supplemental material (p. 2) “Our main hypothesis was that the distance of the participants’ estimate from the true weight would predict their reported surprise, even when being correct was factored in to the model.” I thought the interaction with correctness was the key prediction. This sounds like a main effect which is not what you're predicting.

Further, the title suggests that it is overprecision (which I thought - perhaps incorrectly - always required the use of a confidence interval) was key, not distance, and of course, there is the more standard “how confident are you?” rating. The manuscript doesn’t distinguish between these 3 in any important theoretical way.

It is confusing to include Figure 1 in its current location. It is also referenced following a sentence that doesn’t seem to apply: “The greater the distance between guesses and the truth, the more surprise they reported, β= .77, t(380) = 7.20, p< .001. See Figure 1.” Figure 1 predicts surprise from confidence, not distance of the guess from truth. Further, it says that the values are standardized “by z-scoring within each study” Are these z-scores based on the grand mean or on each individual across the five trials? The latter might cause issues by making a person whose lowest surprise rating was 20 comparable to a person whose lowest surprise rating was, say 60.

The manuscript should probably include inclusion exclusion criteria for mTurk. Number of HITS, % approved, did you restrict identical (mTurk) IDs, did you check for identical GeoLocation (it’s a pain, but particularly if you required a low #HITs and low approval rating (e.g., 90% or lower), you can get a lot of garbage.

6. PLOS authors have the option to publish the peer review history of their article (what does this mean?). If published, this will include your full peer review and any attached files.

Reviewer #1: Yes: Prof. Dr. Theresa Treffers

Reviewer #2: No

---

## [Author Response · Author response to Decision Letter 0]

15 Apr 2020

EDITORIAL DECISION LETTER

Dear Dr. Moore,

Thank you for submitting your manuscript to PLOS ONE. After careful consideration, we feel that it has merit but does not fully meet PLOS ONE’s publication criteria as it currently stands. Therefore, we invite you to submit a revised version of the manuscript that addresses the points raised during the review process.

We would appreciate receiving your revised manuscript by Apr 11 2020 11:59PM. To enhance the reproducibility of your results, we recommend that if applicable you deposit your laboratory protocols in protocols.io, where a protocol can be assigned its own identifier (DOI) such that it can be cited independently in the future. For instructions see: http://journals.plos.org/plosone/s/submission-guidelines#loc-laboratory-protocols

A. We enthusiastically endorse the goal of making project records public. All of the materials, pre-registrations, and data are saved online at the paper’s OSF site: DOI 10.17605/OSF.IO/J5VPE. 

A rebuttal letter that responds to each point raised by the academic editor and reviewer(s). This letter should be uploaded as separate file and labeled 'Response to Reviewers'.

B. We have done so. 

A marked-up copy of your manuscript that highlights changes made to the original version. This file should be uploaded as separate file and labeled 'Revised Manuscript with Track Changes'.

An unmarked version of your revised paper without tracked changes. This file should be uploaded as separate file and labeled 'Manuscript'.

C. We have done so. 

D. Thanks. We would gladly share the article’s review history publicly. 

REVIEWER 1

Review for the paper entitled ‘Overprecision increases subsequent surprise’

Thank you for letting me review this exciting and relevant paper. I believe the authors have investigated a relevant topic in three rigorous studies. Their results show that overprecision increases surprise and reduces subsequent confidence. Although I think this research is important I have some concerns before the results can be published.

Major concerns:

1. My most important concern about the results that are reported in this paper is their external validity. All studies use abstract tasks to measure overprecision and I am not convinced that the results would hold for decisions that have personal consequences for the decision-maker. People in your studies had no incentive to guess the weight of people right. However, people who make overprecise predictions about their future income will experience personal consequences of this judgment. Similarly, managers who have to predict financial losses of their firm will probably experience personal consequences of a wrong judgment. Having said that, I also guess that people will put more thought into judgments that have personal consequences for them which will affect their subsequent surprise. Hence, before I can trust the presented results, I would like to see their external validity and see if this “sensible pattern” holds for (more) realistic settings.

E. We concede that our research contexts might not capture all the high-stakes contexts in which professionals’ confidence judgments might matter. We are entirely sympathetic to this concern, and acknowledge it now in the paper’s discussion. However, no one study can adequately capture the wide variety of decision contexts in which ex-ante confidence judgments might be followed by surprising evidence. Our studies, like other laboratory experiments, provide more control and clear causal identification than apparent similarity to everyday decisions. The obvious benefit is that we can make clear causal claims. The reviewer notes the greater stakes in some everyday decisions, which might increase attention or effort. But we also see the possibility that higher stakes could impede learning when, for instance, belief revision might introduce cognitive dissonance. The paper now suggests that these might be worthy topics for future research. 

2. You state that “When something unexpected happens, it receives more attention and longer gaze […].” As most people are uncertainty averse, they try to avoid or diminish uncertainty or the unexpected. Hence, I would argue that people would like to engage in deliberate consideration, especially for epistemic questions, to exactly avoid to be surprised for situations of personal relevance. In your weight guessing task, some people might not think systematically about cues that would give them the correct answer, but would just make guesses based on unspecific cues. Did you check participants’ time to give their answers or ask them about how they guessed the weight of the strangers?

F. We agree that it is worth considering how research participants’ inattention, laziness, or stupidity might influence the results. In our case, indifferent subjects who were not trying very hard and did not care much would introduce more noise in their responses. This should impair our ability to find any relationship between certainty, accuracy, and surprise. The fact that we do find consistent effects should assuage this concern to some degree. However, prompted by this concern, we went back and looked at how long participants had taken to complete the survey. This measure does not moderate the relationship we observe between confidence, accuracy, and surprise. 

3. Can you really compare the tasks of your studies? The probability distributions in the three tasks used is very different and I guess surprise is lower when you are correct in a coin flip compared to when you are correct in the bingo task or the weight guessing task. Similarly, you measure overprecision AND surprise differently. Why did you use 1-100 in Study 1 and 1 to 7 scales in Study 2 to measure suprise? 

G. We introduced these variations in an attempt to address some of the same generalizability concerns you raised in your point (2) above. We wanted to see that our effect was not dependent on these trivial choices in experimental design. Of course, there are many other features of our research paradigms that we could have varied, but we were limited in the number of variations we could introduce and still preserve enough consistency in our approach that the studies built on each other. 

Why did you not use the surprise subscale from PANAS-X? 

H. We were not aware of the surprise subscale of the PANAS-X. Thank you for bringing it to our attention. We would note that the surprise subscale just includes three items: amazed, surprised, astonished. We assessed the most relevant of the three. 

Related to that, your study design seems quite straightforward; is it possible that participants guessed the purpose of the study and give answers that conform with your hypotheses? Did you ask participants if they knew about the purpose of the study (and control for that)? 

I. It is indeed possible that, participants who guessed our purpose, might have helpfully tried to confirm our hypotheses. It is also possible that devious participants might have intentionally tried to contradict what they thought we expected of them. As it happens, we did have a question at the end of Study 3 that asked, “What do you think this survey was about?” Three (2% of) participants mentioned surprise. Most participants guessed something like, “How we estimate and our confidence.” We doubt that participants’ abilities to intuit our hypotheses exerted a meaningful role on our results. 

Although I am no opponent of using MTurk for data collection, it would be interesting to describe your samples in a bit more detail (gender, age, ethnicity, profession etc.) and possibly also include some control variables in your analyses (at least in Study 1 because it is correlational).

J. We have added some demographic details to the paper for studies 2 and 3, where we collected it. However, we are reluctant to conduct subset analyses or testing for moderation in the absence of a theory predicting its relevance. 

4. I am missing some explanations why surprise does not reduce overprecision. Why should we care that overprecision leads to surprise if we don’t know which consequences surprise has for overprecision or any other (personally relevant) decision? It is possible that not surprise, but another emotion, for example, regret, could be a corrective to reduce overprecision. People may want to avoid feeling regret when finding out that their initial judgment was wrong and regret anticipation could reduce overprecision. This is just an hypothesis, but it would be interesting to test this (in a realistic setting).

K. We do not disagree that this would be worth testing, especially given our failure to find consistent effects of surprise on subsequent confidence. 

Minor comments:

1. A more detailed overview of how overprecision is measured in extant research would be helpful in judging if you apply and compare the relevant measures for overprecision.

L. Thanks for this encouragement. We have added more detail on approaches to measuring overprecision to the paper’s introduction. 

2. Did the photographs show the full body or only faces?

M. The photographs showed the full body, as the revised manuscript now clarifies. All the photos and verbatim copies of the study stimulus materials are available on the study’s OSF web site. 

3. In Study 2, you used 10 questions per domain? Did you (quasi) randomize the questions or blocks of questions?

N. Yes, there were ten bingo ball questions and ten weight-guessing questions. We randomized the order of bingo vs. weight blocks and also randomized question order within each block. The revised manuscript makes this fact clear. 

REVIEWER 2

The idea that greater confidence will produce greater surprise (if incorrect) would seem to be a fairly intuitive thing. The manuscript in its current form doesn’t seem to build a very strong rationale for the research other than nobody else has done it before. 

O. We take this concern seriously; the fact that no one has done it before is a poor reason to conduct a study. We have removed it from our paper. Our paper attempts to honestly present the theoretical questions the motivated our approach. We wanted to investigate an important implication of the research suggesting the ubiquity of overprecision in judgment: frequent surprise. The revised manuscript clarifies this motive. 

Arguably, however, a number of studies in the hindsight bias literature have already demonstrated the relationship being tested in this paper (e.g., Ash, 2009; Choi & Nisbett, 2000; Pezzo, 2003). In these studies participants predict the likelihood of an outcome; those whose predictions were most “off” were found to be most surprised. Inherent in a likelihood judgment is a statement of confidence. If person A predicts a 90% chance that the home team will win, she is more confident than person B who predicts only a 60% chance. As such she will be more surprised if the team loses. The authors’ use of distance of participants’ estimate from the true value as a measure of confidence is consistent with this idea. (Note however, that hindsight bias per se can get convoluted and doesn’t always give the mirror image of surprise). In any case, the manuscript would benefit from an expanded introductory section on the extant literature on surprise, including some of the work by Rainer Reisenzein, mention the hindsight literature, and how this research moves beyond that, and should address theoretical differences between their different measures of confidence.

P. We appreciate this encouragement and have taken you up on your invitation to address the hindsight bias literature in our introduction. 

STUDY 1

Study 1 has some inconsistencies in its data reporting. The manuscript says N = 430, but t-tests give 451 degrees of freedom. The data set provided on the OSF website is a “tall” data set (typical of multilevel modeling) with what should be 5 rows per subject. However, the total number of rows (N = 2240) divided by 5 = 448. A quick check of unique IDs also yields 448. Also, 6 of the 2240 rows have missing data. Finally, the manuscript says that 37% of the guesses were correct (which is consistent with the OSF data set), but the supplemental material says 35%. These inconsistencies need to be cleared up.

Q. We salute your thoroughness! Thank you for caring enough to help us get these details right. We have reviewed and resolved the inconsistencies you identified. This led us to re-run the analyses. This time we did it in R, so that we could share the analysis code (originally the analyses had been conducted in SPSS). These corrections entailed a corrected data file and analysis code which we have posted to the paper’s OSF site. 

A much bigger problem is that the column marked “correct” in the OSF data set is not consistent with the description in the method. The method says that a guess was marked correct if it was within 10lbs of the actual weight. Although the actual weights aren’t shown in the data set, we would expect the range of guesses that are correct (for any given photograph) to be 20lbs (i.e., +/- 10 lbs). Assuming that everyone saw the same photo in a given “round” (I confirmed this by examining the Qualtrics survey) here are the highest and lowest weight guesses for each of the five pictures. Remember, these are just from people marked correct in the OSF data set:

Round 1: Low guess = 150, High guess = 265

Round 2: Low guess = 120, High guess = 210

Round 3: Low guess = 95, High guess = 170

Round 4: Low guess = 120, High guess = 195

Round 5: Low guess = 105, High guess = 180

Clearly something is amiss here. No matter what the true weights are, these all have ranges much greater than 20lbs. As a double check, I inspected everyone, including those who were marked incorrect, and the ranges get even larger (as would be expected), and include some clear typos (e.g., weight estimates below 50lbs). I don’t see anything in the supplemental materials about cleaning any of this data.

I found the true weights in the Qualtrics survey and recalculated the correctness variable. Of the 2240 cases 35.9% were marked correct, but and most important, the original and the newly calculated “correct” columns are completely unrelated (kappa = .09).

The good news is that when you calculate the correlation between surprise and confidence (broken down by correctness, and using the proper “correct” variable), the simple bivariate correlations make a heck of a lot more sense:

r (incorrect) = .40 r (correct) = - 39. Exactly as you would expect.

R. We cannot thank you enough for your thoroughness. By taking the time and doing the hard work to actually dig in to our data, you helped us identify a coding error. We are so grateful to have identified this in the review process. The revised manuscript corrects this error, and we have posted corrected data files to the project’s OSF site. 

STUDIES 2 and 3

The data sets for Studies 2 and 3 seem fine, near as I can tell (although I did not look closely). The method for all three studies should be fleshed out more (e.g., exact wording of the confidence interval question). 

S. We have supplied it.

I found myself often having to refer to the supplemental materials for basic information. The existing method section also has some unnecessary repetition that can be removed to make room for more of these details.

T. In the revision, we have done our best to remove redundancies.

Overall, the presentation of the results is somewhat disorganized. A number of results are presented outside the context of the multilevel modeling and then again repeated within the modeling context (e.g., the main effect of question type). For example, at the top of p.10 a breakdown of the correct x confidence interaction is given before mentioning the interaction (and before mentioning MLM). Then the interaction is mentioned, and then essentially the same breakdown is given (with the exact same results and p-values repeated!). 

U. Thanks for helping us identify these redundancies. We have sought to eliminate them. 

All three studies suffer from this problem. I think the manuscript would benefit from inclusion of a complete table with the results of all analyses in each study

V. Thank you for this suggestion. We have done as you suggest. The table doesn’t quite include every single analysis, since we wanted to honor your encouragement to avoid redundancy. But it includes what we think are the key results. 

Does distance from the true value (or confidence) treated as an individual difference variable (averaged across the 5 pictures for any given person) predict surprise at level 2? I think you can examine cross level interactions as well. Since you are using a random slopes approach (which I assume you tested for, and again should report) you might as well see what predicts the slope for any given person by examining level 2 predictors.

W. Thanks for this suggestion. Aggregated at the individual level, we find that surprise and confidence are unreliably correlated. In Study 1, the correlation is r (443) = .17, p < .001. However, in Study 2, the correlation is r (113) = -.02, p = .83. And in Study 3, the correlation is r (149) = .01, p = .93. We have little reason to think that the results from Study 1 are any more useful or informative than those from Studies 2 and 3. We are reluctant to add these analyses to the paper for several reasons. First, the inconsistency across studies undermines our faith that this is a reliable effect worth report. Second, we are not convinced that they meaningfully contribute to the paper’s main point. Third, as a post-hoc analysis we worry about interpretation and setting the right significance threshold. 

Study 2 uses a very different criterion for correctness (50% confidence interval) vs that of Studies 1 and 3 (“Is the given answer within +/- 10 lbs of the correct answer”), and the implications are never really discussed in the manuscript. In Study 3, the criterion for correct guessing of quarter flips seems a bit arbitrary. A “pilot” test could determine how many flips “off” most students would require before they considered their guess to be wrong. In general I wonder if the participants would agree with the definition of correctness and if there are individual differences. At least discuss this.

X. Thanks for highlighting different correctness criteria across studies. You have correctly identified the fact that most any criteria will, to some extent, be arbitrary. We have added a passage discussing this issue to the paper’s introduction. 

From supplemental material (p. 2) “Our main hypothesis was that the distance of the participants’ estimate from the true weight would predict their reported surprise, even when being correct was factored in to the model.” I thought the interaction with correctness was the key prediction. This sounds like a main effect which is not what you're predicting.

Y. It is not clear to us what document you are referring to here. We could not find the text you quote in the main manuscript, the pre-registration, or Study 2’s results write-up posted on OSF. We pre-registered our plan to test for the interaction in our Hypothesis 2: “When people are wrong, they will report more surprise (ex post) the more confident they were (ex ante); however, when people are right, they will report less surprise (ex post) the more confident they were (ex ante).”

Further, the title suggests that it is overprecision (which I thought - perhaps incorrectly - always required the use of a confidence interval) was key, not distance, and of course, there is the more standard “how confident are you?” rating. The manuscript doesn’t distinguish between these 3 in any important theoretical way.

Z. Thank you for encouraging us to clarify these methodological issues. The revised manuscript does so. 

It is confusing to include Figure 1 in its current location. It is also referenced following a sentence that doesn’t seem to apply: “The greater the distance between guesses and the truth, the more surprise they reported, β= .77, t(380) = 7.20, p< .001. See Figure 1.” Figure 1 predicts surprise from confidence, not distance of the guess from truth. 

AA. Thanks for pointing this out. We have corrected it. 

Further, it says that the values are standardized “by z-scoring within each study” Are these z-scores based on the grand mean or on each individual across the five trials? The latter might cause issues by making a person whose lowest surprise rating was 20 comparable to a person whose lowest surprise rating was, say 60.

BB. Thank you for pointing out this ambiguity. We have clarified now in Figure 1’s caption: we used the grand mean.

The manuscript should probably include inclusion exclusion criteria for mTurk. Number of HITS, % approved, did you restrict identical (mTurk) IDs, did you check for identical GeoLocation (it’s a pain, but particularly if you required a low #HITs and low approval rating (e.g., 90% or lower), you can get a lot of garbage.

CC. Thanks for the encouragement to include this information. We have added what we have. We did not check for redundant geolocations.

---

## [Decision Letter · Decision Letter 1]

22 Apr 2020

PONE-D-19-33924R1

Overprecision Increases Subsequent Surprise

PLOS ONE

Dear Dr. Moore,

Thank you for submitting your revised manuscript to PLOS ONE. Both reviewers were satisfied with your responses and recommended a few additional, minor improvements. I am grateful to both reviewers for their diligant and fast responses. We invite you to submit a revised version of the manuscript that addresses the points raised during the review process.

We would appreciate receiving your revised manuscript by Jun 06 2020 11:59PM. To enhance the reproducibility of your results, we recommend that if applicable you deposit your laboratory protocols in protocols.io, where a protocol can be assigned its own identifier (DOI) such that it can be cited independently in the future. For instructions see: http://journals.plos.org/plosone/s/submission-guidelines#loc-laboratory-protocols

We look forward to receiving your revised manuscript.

Kind regards,

Philipp D. Koellinger, Ph.D.

Academic Editor

PLOS ONE

Reviewers' comments:

Reviewer's Responses to Questions

**Comments to the Author**

1. If the authors have adequately addressed your comments raised in a previous round of review and you feel that this manuscript is now acceptable for publication, you may indicate that here to bypass the “Comments to the Author” section, enter your conflict of interest statement in the “Confidential to Editor” section, and submit your "Accept" recommendation.

Reviewer #1: All comments have been addressed

Reviewer #2: All comments have been addressed

2. Is the manuscript technically sound, and do the data support the conclusions?

Reviewer #1: (No Response)

Reviewer #2: Yes

3. Has the statistical analysis been performed appropriately and rigorously? 

Reviewer #1: Yes

Reviewer #2: Yes

4. Have the authors made all data underlying the findings in their manuscript fully available?

Reviewer #1: Yes

Reviewer #2: Yes

5. Is the manuscript presented in an intelligible fashion and written in standard English?

Reviewer #1: Yes

Reviewer #2: Yes

6. Review Comments to the Author

Reviewer #1: Thank you very much for your thorough answers to my comments and the according revisions you have made to the paper. I still have a few comments left that will hopefully further improve the paper:

- For Study 1, you report an attention test that was done right after participants have received the instructions. I would have wished that the attention test was rather later in your study to check if participants were attentive throughout the study. What you describe as attention test appears more like a check if participants have read and understood the instructions which is of course also important.

- From your answer J. I take it that you didn’t collect demographic details from your sample in Study 1. If so, please state this in the Method for Study 1.

- For Study 2, you seem to have collected demographic information for your sample, but you don’t report the gender of the participants. Please check if you really did not collect information about participants’ gender.

- In Table 1, please report p-values for the correlation coefficients. I also think the position of Figure 1 and Table 1 within the text for Study 1 is a bit out of place. I would suggest to either place it at the end of the chapter “present research” or after having presented all three studies in detail.

- In your results and discussion for Study 3 on p. 17 you write: “Were participants as surprised as they predicted they should be? In order to test this, we employed two independent-samples t-tests to account for the repeated measures design, comparing the predicted surprise for a correct answer to a subsequent correct answer’s surprise, and comparing the predicted surprise for an incorrect answer to a subsequent incorrect answer’s surprise.” Shouldn’t this be a DEPENDENT t-test?

- In your general discussion on p. 21 about the “utility of surprise”, I’d like to see more concrete links to current literature and more precise suggestions for future research, e.g., which specific moderators may be promising to include in future studies?

Reviewer #2: The manuscript is much improved. The increased detail in the method is greatly appreciated. The inclusion of the surprise and hindsight sections really help too. Except for a few very minor things, I think the manuscript is ready to go, and I look forward to citing it in the future.

Minor Things:

1. Table 1 is MUCH better. You might want to change Study 1 to be consistent with the other two "Weights - right" and "Weights - wrong" instead of "When right" and "When wrong"

2. I'd love to see more of a rationale for including the prediction of surprise in the intro to Study 3

3. I think a bit more distinction between (confidence predicting surprise) and (surprise at t-1 predicting confidence) is in order in the intro or general discussion. The latter sort of pops up out of nowhere, and then quickly disappears. Related to this, the last sentence and second to last sentence before the general discussion seem a bit at odds with one another. Maybe replacing "In short" with "Nevertheless" would do the trick?

4. Study 3 doesn't appear to test for the condition x correct interaction, although I may have missed it.

5. My reference to the supplemental material (your reply "Y") was incorrect. It was in Study 1, not Study 2 that I saw this passage. I think the same question applies though.

That's it!

7. PLOS authors have the option to publish the peer review history of their article (what does this mean?). If published, this will include your full peer review and any attached files.

Reviewer #1: Yes: Prof. Dr. Theresa Treffers

Reviewer #2: No

---

## [Author Response · Author response to Decision Letter 1]

20 May 2020

Reviewer #1

Thank you very much for your thorough answers to my comments and the according revisions you have made to the paper. I still have a few comments left that will hopefully further improve the paper:

- For Study 1, you report an attention test that was done right after participants have received the instructions. I would have wished that the attention test was rather later in your study to check if participants were attentive throughout the study. What you describe as attention test appears more like a check if participants have read and understood the instructions which is of course also important.

A. Thanks for noting this issue. Obviously, there are tradeoffs here. One reason to put the attention check before any experimental treatments or manipulations is to reduce concerns with differential attrition. That is, if the attention check comes late in the study, then it is possible that it induces differential drop-out rates correlated with the independent variable and therefore impairing the exogeneity of the experimental manipulation and undermining random assignment to condition. 

- From your answer J. I take it that you didn’t collect demographic details from your sample in Study 1. If so, please state this in the Method for Study 1.

B. Correct. We have done as you suggest. 

- For Study 2, you seem to have collected demographic information for your sample, but you don’t report the gender of the participants. Please check if you really did not collect information about participants’ gender.

C. We did indeed collect gender data in Study 2. We have added the gender breakdown to the study’s reported methods. 

- In Table 1, please report p-values for the correlation coefficients. I also think the position of Figure 1 and Table 1 within the text for Study 1 is a bit out of place. I would suggest to either place it at the end of the chapter “present research” or after having presented all three studies in detail.

D. We have moved Figure 1 and Table 1 to after the presentation of Study 3 and added asterisks reflecting the statistical significance of correlation coefficients. 

- In your results and discussion for Study 3 on p. 17 you write: “Were participants as surprised as they predicted they should be? In order to test this, we employed two independent-samples t-tests to account for the repeated measures design, comparing the predicted surprise for a correct answer to a subsequent correct answer’s surprise, and comparing the predicted surprise for an incorrect answer to a subsequent incorrect answer’s surprise.” Shouldn’t this be a DEPENDENT t-test?

E. Good catch! Those are, in fact, paired t-tests, as the revised manuscript now makes clear. 

- In your general discussion on p. 21 about the “utility of surprise”, I’d like to see more concrete links to current literature and more precise suggestions for future research, e.g., which specific moderators may be promising to include in future studies?

F. Thanks for this encouragement. We have elaborated on opportunities for future research. 

 

Reviewer #2

The manuscript is much improved. The increased detail in the method is greatly appreciated. The inclusion of the surprise and hindsight sections really help too. Except for a few very minor things, I think the manuscript is ready to go, and I look forward to citing it in the future.

Minor Things:

1. Table 1 is MUCH better. You might want to change Study 1 to be consistent with the other two "Weights - right" and "Weights - wrong" instead of "When right" and "When wrong"

G. Good suggestion. We have done so. 

2. I'd love to see more of a rationale for including the prediction of surprise in the intro to Study 3

H. That’s easy. We have elaborated on our reasoning in Study 3’s method section. 

3. I think a bit more distinction between (confidence predicting surprise) and (surprise at t-1 predicting confidence) is in order in the intro or general discussion. The latter sort of pops up out of nowhere, and then quickly disappears. 

I. Thanks for this suggestion. We have highlighted the value of this test in the introduction where we describe the motivations behind each study’s design. 

Related to this, the last sentence and second to last sentence before the general discussion seem a bit at odds with one another. Maybe replacing "In short" with "Nevertheless" would do the trick?

J. Done. 

4. Study 3 doesn't appear to test for the condition x correct interaction, although I may have missed it.

K. Thanks for asking about the interaction in Study 3. Given the results of Studies 1 and 2, and given Study 3’s replicated reversal, in which the correlation between ex-ante confidence and ex-post surprise flips from -.27 (p < .001) for correct predictions to .11(p < .001) for incorrect predictions, we believe a test of the interaction is superfluous. 

5. My reference to the supplemental material (your reply "Y") was incorrect. It was in Study 1, not Study 2 that I saw this passage. I think the same question applies though.

L. Thanks for encouraging us to test for the interaction in Study 1. It is indeed significant, and we now report it.

---

## [Decision Letter · Decision Letter 2]

2 Jun 2020

Overprecision Increases Subsequent Surprise

PONE-D-19-33924R2

Dear Dr. Moore,

We’re pleased to inform you that your manuscript has been judged scientifically suitable for publication and will be formally accepted for publication once it meets all outstanding technical requirements.

Kind regards,

Philipp D. Koellinger, Ph.D.

Academic Editor

PLOS ONE

Additional Editor Comments (optional):

Reviewers' comments:

Reviewer's Responses to Questions

**Comments to the Author**

1. If the authors have adequately addressed your comments raised in a previous round of review and you feel that this manuscript is now acceptable for publication, you may indicate that here to bypass the “Comments to the Author” section, enter your conflict of interest statement in the “Confidential to Editor” section, and submit your "Accept" recommendation.

Reviewer #1: All comments have been addressed

Reviewer #2: All comments have been addressed

2. Is the manuscript technically sound, and do the data support the conclusions?

Reviewer #1: Yes

Reviewer #2: Yes

3. Has the statistical analysis been performed appropriately and rigorously? 

Reviewer #1: Yes

Reviewer #2: Yes

4. Have the authors made all data underlying the findings in their manuscript fully available?

Reviewer #1: Yes

Reviewer #2: Yes

5. Is the manuscript presented in an intelligible fashion and written in standard English?

Reviewer #1: Yes

Reviewer #2: Yes

6. Review Comments to the Author

Reviewer #1: Thank you for addressing my final comments. I like the final paper very much and believe it makes an important contribution to our knowledge about the consequences of overconfidence.

Reviewer #2: I did find one typo:

Study 3 invites participants to report how confident they are that the truth will be close **TO** their best guess.

Otherwise, the manuscript appears to be in good shape.

7. PLOS authors have the option to publish the peer review history of their article (what does this mean?). If published, this will include your full peer review and any attached files.

Reviewer #1: Yes: Prof. Dr. Theresa Treffers

Reviewer #2: Yes: Mark V. Pezzo

---

## [Editor Report · Acceptance letter]

17 Jun 2020

PONE-D-19-33924R2 

Overprecision Increases Subsequent Surprise 

Dear Dr. Moore:

I'm pleased to inform you that your manuscript has been deemed suitable for publication in PLOS ONE. Congratulations! Your manuscript is now with our production department. 

Kind regards, 

on behalf of

Dr. Philipp D. Koellinger 

Academic Editor

PLOS ONE